# Pd/Attapulgite Core–Shell Structured Catalytic Combustion Gas Sensor for Highly Sensitive Real-Time Methane Detection

**DOI:** 10.3390/s25164950

**Published:** 2025-08-10

**Authors:** Shuo Cao, Shuang Pang, Zishuai Zhang, Lulu Feng, Chong Zhang, Jiahao Lin, Zhiyu Liu, Yifei Sun, Shiyu Wang, Zhenning Lou

**Affiliations:** 1College of Physics, Liaoning University, Shenyang 110036, China; shuocao@lnu.edu.cn (S.C.); pangshuangemail@163.com (S.P.); z1061897905@163.com (Z.Z.); ll110810278@163.com (L.F.); 18865405741@163.com (C.Z.); l2579361117@163.com (Z.L.); 2College of Chemistry, Liaoning University, Shenyang 110036, China; lamchem@163.com (J.L.); louzhenning@lnu.edu.cn (Z.L.); 3College of Information Engineering, Shenyang University of Chemical Technology, Shenyang 110142, China; sunyifei@syuct.edu.cn

**Keywords:** catalytic combustion, gas sensors, attapulgite, core–shell structure, methane

## Abstract

Catalytic combustion gas sensors are critical for safety and environmental monitoring, yet face persistent challenges in sensitivity and detection limits amid expanding market demands. This study innovatively employs attapulgite as a support material functionalized with noble metal catalyst Pd to fabricate a low-cost, high-sensitivity sensor. Characterization (SEM/EDS) confirms a unique Pd/attapulgite core–shell structure with engineered Pd gradient distribution (7.5–75.8 wt% from core to surface). The sensor achieves methane catalytic combustion below 300 °C, delivering 0.7 µV/ppm sensitivity and ~36 ppm detection limit. Reaction kinetics follow the Eley–Rideal mechanism, with voltage difference (ΔU) versus methane concentration (C) conforming to the Langmuir equation ($\Delta U=\frac{U_{max}\cdot K\cdot C}{1+K\cdot C}$, R^2^ > 0.99, $U_{max}$ = 41.80 mV). Cost-effective fabrication and exceptional performance underscore its potential for practical deployment in industrial, residential, and environmental safety monitoring.

## 1. Introduction

Driven by heightened global safety awareness, the advancement of intelligent technologies, and growth in the energy sector, the catalytic combustion gas sensor market continues to exhibit steady expansion [1,2,3]. Industrial and household environments face significant safety risks due to abundant flammable gases (e.g., methane) and volatile organic compounds (VOCs). Gas leakage incidents can readily trigger fires or explosions, endangering lives and causing substantial economic losses. As a critical technology for industrial safety and residential protection [4,5,6], catalytic combustion gas sensors serve as core monitoring devices for combustible gas detection due to their simple structure, low cost, and high sensitivity, holding pivotal roles in industrial safety systems and residential gas alarms [7]. Despite their widespread applications, these sensors still face performance challenges, including sensitivity and detection limits in complex real-world environments [8].

Current catalytic combustion sensors primarily comprise a detection element (“black bead”) coated with noble metal catalysts and a catalyst-free compensation element (“white bead”), connected via a Wheatstone bridge (Figure 1) [9,10,11]. When combustible gas contacts the detection element, catalysts (e.g., Pd, Pt) promote methane oxidation with oxygen, releasing heat that increases the resistance of the internal platinum wire. The compensation element counters ambient temperature fluctuations. Gas concentration is quantified by comparing the potential difference between the two elements. Noble metals (Pd, Pt, Rh, and Au) and their oxides are typically used as catalysts owing to their superior activity. However, metals beyond Pt/Pd suffer from high volatility, oxidation susceptibility, and limited supply, restricting their practical application. Thus, Pt and Pd remain the most widely adopted catalysts for methane combustion [9].

In addition, support materials critically determine sensor performance by fulfilling three essential functions: providing porous architectures to achieve highly dispersed anchoring of noble metal catalysts, thus maximizing active sites and enhancing sensitivity; ensuring long-term thermal stability to prevent catalyst agglomeration during high-temperature operation; and potentially optimizing catalytic activity through metal–support interactions. Consequently, supports exhibiting high specific surface area, excellent thermal stability, tailored pore structures, and chemical inertness form the foundation for high-performance catalytic combustion gas sensors. Currently, conventional supports (e.g., cordierite/Al_2_O_3_ ceramics, Si-MEMS [12,13,14]) ace several intrinsic limitations: high manufacturing cost, low specific surface area (<50 m^2^/g), and insufficient functional groups for catalyst anchoring [10]. Attapulgite (palygorskite), a one-dimensional hydrous layer-chain magnesium silicate mineral, exhibits excellent physicochemical properties including hierarchical porosity, a large specific surface area, ion adsorption capacity, hydrophilicity, exceptional thermal stability (>500 °C), abundant surface hydroxyl groups, and non-toxicity [11,15,16,17]. As a naturally abundant, low-cost nanoclay, it has found widespread application in industrial sectors such as catalysis, oil/water separation, ion adsorption, and corrosion protection. The recent research shows that attapulgite has been utilized in humidity sensing [18] and halloysite nanotubes (HNTs, a structural analogue) in room-temperature NH_3_ sensors [19], but its application in high-temperature catalytic combustion gas sensors remains unexplored. To address these limitations of the conventional supports in the catalytic combustion gas sensors field, we pioneer attapulgite as a novel catalytic support, forming a novel Pd/attapulgite core–shell structured catalytic combustion gas sensor for highly sensitive real-time methane detection.

This work reports a novel catalytic combustion gas sensor based on an attapulgite support. Pretreatments (thermal/acid etching) increased carrier porosity and specific surface area, enhancing gas adsorption capacity [15]. Uniform Pd deposition was achieved via palladium salt solution infiltration, facilitating confined growth within nanopores. The Pd/attapulgite core–shell structure was verified by SEM and XPS characterization. Methane sensing tests demonstrated outstanding sensitivity (0.7 µV/ppm) and a limit of detection (36 ppm). This sensor exhibits significant potential for safety monitoring and environmental surveillance applications.

## 2. Materials and Methods

### 2.1. Sensor Architecture

The sensor integrates three core components: a detection element (black bead) that catalyzes methane oxidation and transduces chemical energy into platinum wire resistance changes; a compensation element (white bead) counteracting ambient temperature variations; and an encapsulation metal mesh serving as both a physical barrier against particulate damage and a thermal insulator to minimize heat dissipation.

The detection and compensation elements are interconnected via a Wheatstone bridge circuit. Surface Pd catalysts on the detection element drive exothermic methane combustion, converting chemical signals (CH_4_ concentration) into thermal energy. Through thermal conduction, this elevates the platinum wire’s temperature, increasing its resistance to generate electrical signals. Methane concentration quantification is achieved by measuring the resultant potential difference across the bridge. The fabrication workflow and structural configuration are illustrated in Figure 1.

### 2.2. Fabrication of Detection/Compensation Elements

The attapulgite was purchased from Tianke Mineral Co., Ltd. (Mingguang, China) and used as received without further purification. The platinum wire (Pt, 99.99%) was purchased from Boyan Technology Co., Ltd. (Taizhou, China). Palladium nitrate (Pd(NO_3_)_2_, 18.09 wt% in H_2_O) was provided by Shanghai Macklin Biochemical Technology Co., Ltd. (Shanghai, China). Hydrochloric acid (HCl, 36~38%) and ammonium persulfate ((NH_4_)_2_S_2_O_8_, APS < 99.0%) were purchased from Shanghai Aladdin Biochemical Technology Co., Ltd. (Shanghai, China).

Platinum coil preparation: An 80 μm platinum wire was tightly coiled around a 200 μm copper mandrel, annealed (500 °C, 2 h, Ar atmosphere), etched in ammonium persulfate solution to remove copper, rinsed with deionized (DI) water, and dried. Attapulgite sphere assembly: 10 g of attapulgite was vortex-oscillated in 30 mL DI water, with 8 μL dispersion drop-casted twice onto platinum coils to form uniform spheres, air-dried at 25 °C. Support functionalization: Spheres were thermally treated in a tube furnace (500 °C, 2 h, Ar atmosphere), and then cycled twice through 0.1 M HCl immersion/DI water rinsing/drying → white bead. Pd catalyst loading: White beads were impregnated with Pd(NO_3_)_2_ solution, air-dried, thermally reduced (500 °C, 2 h, Ar atmosphere) → black bead, and then labeled by Pd concentration: P1 (2 wt%), P2 (2.5 wt%), P3 (3 wt%), P4 (3.5 wt%), and P5 (4 wt%).

### 2.3. Sensor Bonding and Encapsulation

Black and white beads were resistance-welded onto identical dual-pin headers—forming the detection element (black bead) and compensation element (white bead)—with matched resistances (±5%) verified to ensure Wheatstone bridge equilibrium. The elements were encapsulated within a metallurgical powder mesh using epoxy resin, followed by aging at 150 °C for 24 h to stabilize performance, thus completing sensor assembly.

### 2.4. Characterization

Surface/internal morphology of black beads was characterized by scanning electron microscopy (SEM, Regulus 8100, Hitachi, Beijing, China). Elemental distribution and composition were determined via energy-dispersive X-ray spectroscopy (EDS, Ultim Max 65, Oxford Instrument Technology Co., Ltd., Shanghai, China). Crystal structures of black/white beads were analyzed using X-ray diffraction (XRD, TD-3500X, Dandong Tongda Technology Co., Ltd., Dandong, China). Specific surface area and pore size distribution of white beads were measured via N_2_ physisorption (BET, PhysiChem iPore 1000, PhysiChem Instruments Ltd., Beijing, China). Functional groups were identified through Fourier-transform infrared spectroscopy (FTIR, Thermo Nicolet 5700, Thermo Fisher Scientific, Waltham, MA, USA). Elemental composition and chemical states of black beads were investigated by X-ray photoelectron spectroscopy (XPS, Thermo Fisher 250xi, Thermo Fisher Scientific, Waltham, MA, USA). Operating temperature was monitored using infrared thermography (TOPLIA TMi120s, Guangxi Nanning Zhijiewei Instrument Co., Ltd, Nanning, China). The gas chromatograph (GC 5190 F, Anhui Chromatograph Co., Ltd., Anhui, China) was used to analyze the composition and content in this study.

### 2.5. Performance Testing

Sensor performance was evaluated using the JF02F Gas Sensing Analysis System (GuiYanJinFeng Corp, Tech, LTD, Kunming, China), which provides ppm-level methane concentration (C) control and real-time potential difference (ΔU) monitoring. Prior to each test, the chamber was purged with dry air while the operating temperature was regulated via input voltage (1.7–2.4 V); this enabled systematic optimization of operating temperature (296 °C via Arrhenius analysis), whereupon optimized sensors underwent comprehensive evaluation of sensitivity, repeatability, and long-term stability.

## 3. Results and Discussion

### 3.1. Structural and Elemental Analysis

The procedure in Section 2.1 and Section 2.3 was followed to fabricate the catalytic combustion gas sensor. Structural characterization of the fabricated sensor revealed that the surface morphology of its black bead exhibited ridge-like structures with submicron-scale cracks (~500 nm width, Figure 2a). The internal architecture further showed a 3D network of interlocked nanorods (Figure 2b), which is consistent with attapulgite’s rod-like nanostructure [11]. Elemental analysis via EDS mapping (Figure 2c) confirmed dominant components (Si, Al, Mg, O, and exogenous Pd) alongside trace impurities (Ca, Fe), aligning with attapulgite’s typical composition [17].

Crucially, Pd displayed a gradient distribution: surface concentration reached 75.8 wt%, contrasting sharply with the core (7.5 wt%). Pd uniformly coated the bead surface (Figure 2a inset), while sparse internal dispersion (Figure 2c) evidenced a core–shell architecture—an attapulgite core with a Pd-rich shell.

XRD spectrum (Figure 2d) revealed shared diffraction peaks at 20.3° (attapulgite [110] plane) and 26.8° (quartz [101]) [16,20] for both beads, confirming attapulgite as the dominant carrier phase with trace quartz impurities, while the black bead exclusively exhibited a 40.0° peak corresponding to metallic Pd [111] [17], demonstrating Pd^0^ nanoparticles predominantly loaded on the attapulgite surface.

The N_2_ adsorption–desorption isotherm of white beads (attapulgite support) exhibited a characteristic Type IV curve (Figure 3a–c) [11,17,21,22], confirming their porous nature. Quantitative analysis (Table 1) revealed raw material pore volume = 0.33 cm^3^·g^−1^, surface area = 138.98 m^2^·g^−1^, and dominant pore size = 0.63 nm. After thermal treatment, pore volume increased to 0.39 cm^3^·g^−1^ while surface area decreased to 91.39 m^2^·g^−1^ with enlarged dominant pore size (18.72 nm). Acid etching further enhanced pore volume to 0.47 cm^3^·g^−1^, recovered surface area to 139.98 m^2^·g^−1^, and maximized pore size (21.12 nm). Pore size distribution (Figure 3d) demonstrated synergistic formation of a dual-model pore system—preserved 0.4–0.6 nm micropores alongside constructed 8–23 nm mesopores—which was attributed to thermal-induced microporous collapse/reorganization into uniform mesopores (enhancing structural stability despite reduced surface area) while acid dissolution of Fe/Ca oxide impurities cleared and expanded channels (restoring surface area and boosting pore volume). Collectively, this synergistic processing optimizes the carrier by (i) establishing micro-mesoporous architecture (micropores for confinement, mesopores for mass transport); (ii) enhancing pore stability/volume; and (iii) providing high-surface-area hierarchical channels as ideal anchoring sites for active components, potentially elevating catalytic performance.

FTIR analysis of black (Pd/attapulgite) and white (attapulgite) beads revealed three critical differences (Figure 4a, Table 2): (i) significant attenuation of the O-H stretching peak at 3454 cm^−1^ in black beads, suggesting Pd-coordinated hydroxyl formation (Pd-OH) that consumed surface -OH groups during high-temperature Ar treatment; (ii) a new peak at 1433 cm^−1^ attributed to C-O asymmetric stretching in carbonate/carboxylate species, indicating Pd-catalyzed reduction of adsorbed CO_2_; (iii) a 9 cm^−1^ redshift of the Si-O framework vibration (1025 → 1016 cm^−1^), implying Pd-induced structural modification (e.g., Si-O-Pd bond formation) or lattice strain. These results confirm Pd/attapulgite interactions via coordination and structural alteration.

Complementary XPS analysis (Figure 4b) detected Si, O, Al, Pd, Mg, and trace Ca/Fe—consistent with EDS—while Pd 3d_5/2_ deconvolution (Figure 4c) showed peaks at 336.5 eV (Pd^0^) and 338.1 eV (Pd^2+^). The presence of Pd^0^ confirms effective reduction of Pd(NO_3_)_2_ in Ar, whereas Pd^2+^ may originate from Si-O-Pd bonds hypothesized by FTIR, demonstrating co-existing reduced (Pd^0^) and oxidized (Pd^2+^) species on the carrier surface.

### 3.2. Working Temperature

The catalytic efficiency of combustion sensors exhibits strong temperature dependence: sub-ignition temperatures impede methane oxidation kinetics, while excessive heat accelerates catalyst sintering. To optimize the efficiency–stability trade-off, sensor temperature (T) was precisely controlled via operating voltage regulation (U = 1.7–2.4 V) using the JF02F system, with infrared thermography (Figure 5e–l) providing real-time thermal profiling to establish T-U correlations (Figure 5d). At 2000 ppm CH_4_, the potential difference response (ΔU) displayed sigmoidal growth against temperature (Figure 5a,b), progressing through three distinct regimes: (i) gradual increase in low-T zone (A–B); (ii) exponential surge beyond the inflection point (B, ignition temperature); and (iii) response saturation in the high-T region (C–D) [23]. Arrhenius analysis (Equation (1)) of the kinetically controlled B-C interval confirmed exponential ΔU−T dependence (R^2^ > 0.97, E_a_ = 78.7 kJ·mol^−1^; Figure 5a–c), which is mathematically described by Equation (2), with the results validating exceptional thermal sensitivity and the critical role of temperature modulation in enhancing catalytic response.(1)$$\ln\left(\Delta U\right)=\ln\left(A\right)-\frac{E_{a}}{R}\times \frac{1}{T}$$(2)$$\Delta U=Ae^{-\frac{Ea}{RT}}$$
where ΔU represents the potential difference response (mV), A represents the preexponential factor, E_a_ represents the activation energy (kJ·mol^−1^), R represents the molar gas constant (8.314 J·mol^−1^·K^−1^), and T represents the temperature (K).

To balance catalytic efficiency and stability, 296 °C (2.1 V)—the midpoint of region B–C—was selected as the optimal operating temperature. This temperature resides within the exponential response zone (B–C), where Arrhenius fitting (E_a_ = 78.7 kJ·mol^−1^, R^2^ > 0.97) confirms extreme thermal sensitivity of reaction kinetics. Critically, 296 °C lies significantly below the Pd sintering threshold (>350 °C) [24,25,26], effectively suppressing thermal deactivation. Operating in this moderate temperature range for methane combustion ensures: (1) high catalytic efficiency; (2) power reduction; and (3) minimized thermal runaway risks—achieving synergistic optimization of efficiency, durability, and energy efficiency.

### 3.3. Ambient Temperature Effects

As Section 3.2 shows, the sensing performance of the catalytic combustion gas sensor strongly relates to the working temperature. Thus, it is intrinsically sensitive to thermal perturbations, where ambient temperature fluctuations may alter the operating temperature of the detection element. To evaluate the extent of ambient temperature disturbance, we systematically characterized the thermal and electrical stability of our Pd/attapulgite sensor under varying ambient temperature (25–60 °C) at a stable operating voltage (1.05 V).

Figure 6a displays the voltage–current/working temperature relationship for the detection element (black bead) measured at 25 °C (ambient temperature). This curve exhibits a distinct positive dependence of resistance on voltage, indicative of the platinum wire’s intrinsic positive temperature coefficient (PTC) behavior. Figure 6b presents corresponding measurements under varying ambient temperatures (25–60 °C). Critically, these curves demonstrate near-perfect overlap, signifying that changes in ambient temperature exert negligible influence (<0.05% resistance deviation) on both the operating temperature and electrical resistance of the sensor within this range.

Figure 6c,d show that the working temperature and resistance of the detection element (black bead) under the stable voltage (1.05 V) across ambient temperatures range from 25 to 60 °C. Critically, the resistance of the detection element remained stable at 1.38 Ω (Figure 6c), while the working temperature remained 296 °C (Figure 6d), demonstrating exceptional thermal and electrical stability despite ambient temperature variations. These results collectively demonstrate a concentration-dependent rather than an ambient-dependent response; thus, this sensor achieves ambient temperature-decoupling, ensuring working temperature stability within 25–60 °C.

### 3.4. Electrical Signal Response

As depicted in Figure 7a (Pd loading: P1 = 2 wt% to P5 = 4 wt%), all sensors demonstrated robust responses, yet the maximum differential voltage followed a volcano-shaped trend versus Pd content, peaking at P3 (3 wt%). At 10,000 ppm methane, P3 achieved a 6.94 mV response, which is obviously higher than other variants (Figure 7b). This optimization threshold is attributed to insufficient active sites below 3 wt% Pd loading versus nanoparticle aggregation-induced site blockage beyond 3 wt%.

P1–P3 exhibit progressively enhanced activity due to increased active site density; the performance decline in P4/P5 stems from Pd overloading-induced nanoparticle aggregation and pore blockage (Appendix A). Specifically, excessive Pd in P5 (4 wt%) leads to a large aggregation of nanoparticles, thus occluding mesopores (Figure 3d). This result may cause a significant decrease in porosity, impeding gas diffusion and heat transfer. This microstructural degradation explains the non-applicability of rate-limiting reaction models to P4/P5, as their kinetics are governed by mass transfer and heat transfer limitations rather than intrinsic catalytic activity. This demonstrates that the sensors’ performances are both governed by support microstructure and Pd spatial distribution, which are key factors requiring deliberate engineering for sensor optimization.

All sensors adhered to the Langmuir adsorption model (Equation (3), R^2^ > 0.99; Figure 7c). The optimal P3 sample yielded fitting parameters of $U_{max}$ = 41.8 mV (theoretical maximum response) and K = 2.0 × 10^−5^ (adsorption equilibrium constant), confirming that 3 wt% Pd loading maximizes methane adsorption–catalysis efficiency:(3)$$\Delta U=\frac{U_{max}\cdot K\cdot C}{1+K\cdot C}$$
where ΔU represents the potential difference response (mV), $U_{max}$ represents the theoretical maximum response (mV), K represents the adsorption equilibrium constant, and C represents the methane concentration (ppm).

P3’s dynamic response across 900–10,000 ppm CH_4_ (Figure 7d) demonstrated a maximum ΔU of 0.8 mV at 900 ppm with rapid kinetics: response time τ_res_ = 65.5 s (T_90_ = 65.5 s, T_70_ = 36 s, T_90_/T_70_ ≈ 1.8, Equation (4)) and recovery time τ_recov_ = 163.5 s. Response–recovery profiling (Figure 7e) and fitting analysis (Figure 7f) further revealed the following: low-concentration linear sensitivity = 0.7 µV·ppm^−1^ (Equation (5)), Langmuir-modeled maximum response $U_{max}$ = 41.8 mV, and the detection limit LOD = 36.14 ppm (3σ noise = 0.03 mV, Equation (6)), which is significantly below methane’s explosion threshold (1% LEL = 50 ppm), thus ensuring reliable early-warning capability.τ res = T_90_ − T_0_  τ recov = T_m_ − T_n_(4)
where τ res/τ recov represents the response/recovery time (s), T_90_ indicates that the response value is 90% of the maximum response value (s), T_0_ represents the moment when methane begins to be introduced (s), T_m_ represents the moment to end the introduction of methane (s), and T_n_ represents the moment when the response value returns to the initial value (s).(5)$$Sn=\frac{△y}{△x}$$
where S_n_ represents the sensitivity of the methane sensor (mV/ppm), ∆y represents the amount of change in the voltage difference (mV), and ∆x represents the amount of change in methane concentration (ppm).(6)$$LOD=\frac{3R}{\underset{c\to 0}{\lim}(\frac{dU}{dC})}$$
where LOD represents the limit of detection, $\underset{c\to 0}{\lim}(\frac{dU}{dC})$ represents the initial slope of the Langmuir equation, and R represents the noise (mV).

In addition, we experimentally validated the thermal balance principle governing sensor operation. As shown in Appendix A, the steady-state temperature of the sensing element exhibits strictly linear proportionality to methane concentration (0–5000 ppm). This linear correlation (R^2^ > 0.99) confirms that within this operational window, heat generation from catalytic combustion is precisely balanced by heat dissipation, satisfying the fundamental condition for temperature–concentration linearity established by Nagai et al. [14].

### 3.5. Response Rate

For practical quantification in catalytic combustion gas sensors, the response rate (r) is approximated using the T90 metric (Equation (7)): the voltage difference at 90% saturation (ΔU_T90_) divided by the corresponding response time (T_90_). As summarized in Table 3, these normalized response rates for sensors P1–P5 are 61.59%, 82.70%, 100%, 43.94%, and 41.18%, respectively.(7)$$r=\frac{d{n_{CH}}_{4}}{dT}\propto \frac{dQ}{dT}\propto \frac{△U}{△T}$$(8)$$X=\frac{r_{example}}{r_{p3}}$$
where r represents the response rate (mol·s^−1^), ${n_{CH}}_{4}$ represents the molar amount of methane (mol), Q represents the heat released by the reaction (J), ∆U represents the voltage difference value (mV), T represents the response time (S), and X represents the normalized response rate (%).

Table 3 reveals a distinct volcano-shaped relationship between Pd content and sensor response, with the maximum response rate (28.9 µV/s) at the P3 composition. This optimal performance probably arises from two synergistic factors: optimal Pd dispersion that maximizes the catalytically active surface area and a surface enrichment gradient (75.8 wt% Pd concentration, Figure 2c) that enhances gas accessibility to active sites.

The suboptimal performance at other compositions suggests critical composition–function relationships: P1 exhibits limited activity, consistent with insufficient Pd availability for catalytic processes. P5 shows a 58.82% reduction in normalized activity despite higher Pd content, likely attributable to excessive Pd accumulation at the surface. This agglomeration appears to block mesopores, which may impair heat/mass transfer efficiency and consequently reduce catalytic activity.

Previous studies on palladium (Pd)-doped gas sensors have confirmed that Pd doping significantly enhances sensor response, exhibiting a characteristic volcano-shaped relationship between response magnitude and Pd concentration [27,28,29,30]: (i) for methanol detection, Qiang Hu et al. demonstrated that 3% Pd-doped CeO_2_ nanofibers (via electrospinning) achieved a maximal response of 6.95 to 100 ppm methanol at 200 °C [27]; (ii) in ethanol sensing, Tianjiao Qi et al. reported that 5 wt% Pd-decorated ZnO delivered a peak response of 311 to 50 ppm ethanol at 250 °C, leveraging Pd-induced catalytic activation [28]; (iii) for NO_x_ detection, Shobha N. Birajdar et al. identified 1 wt% Pd@V_2_O_5_ as the optimal formulation, exhibiting superior selectivity toward NO_x_ over interferents (e.g., SO_x_, NH_3_, and VOCs) [29]; and (iv) regarding n-butanol sensing, Zhiqiang Yao et al. revealed that 3 wt% Pd-doped SnO_2_ nanoparticles maximized response [30]. Collectively, these studies indicate that a Pd doping range of 1–5% constitutes the optimal window for response rate. This consistent trend aligns closely with our experimental findings.

### 3.6. Catalytic Combustion Reaction Efficiency

Figure 8a presents a representative gas chromatography (GC) analysis of exhaust gases from the sensor chamber for the P3 sample. Distinct CH_4_ and CO_2_ peaks confirm catalytic methane oxidation. Catalytic efficiency (η) was calculated from their GC peak area ratio (Figure 8a, Equation (9)):(9)$$\eta =\frac{{Sn}_{{CO}_{2}}}{{S_{{CO}_{2}}+S}_{{CH}_{4}}}\times 100\%$$
where η represents the catalytic efficiency (%), $S_{{CO}_{2}}$ represents the characteristic peak area of carbon dioxide, and $S_{{CH}_{4}}$ represents the characteristic peak area of methane.

Table 4 summarizes η values across Pd doping concentrations. Catalytic efficiency follows a distinct volcano-shaped dependence: increasing from P1 (2 wt%, η = 11.20%) to P3 (3 wt%, η = 12.18%), then decreasing to P5 (4 wt%, η = 11.28%), with peak efficiency at P3. This efficiency trend directly correlates with the electrical response (Section 3.4) and response rate (Section 3.5) volcano plots, confirming that Pd concentration critically governs catalytic performance.

In summary, catalytic efficiency measurements provide independent validation of the volcano relationship between Pd concentration and sensing performance. The P3 sensor (3 wt% Pd) achieves optimal efficiency by balancing active site density against agglomeration constraints. Consistency between reaction efficiency (Section 3.6), response rate (Section 3.5), and electrical signal response (Section 3.4) robustly confirms the performance maximum at 3 wt% Pd doping.

### 3.7. Repeatability and Specificity

The sensor demonstrated exceptional repeatability and stability across rigorous testing protocols: (1) 100 response cycles at 2000 ppm CH_4_ showed minimal response deviation (range = 0.01 mV, Figure 9a); (2) after 40-day ambient exposure, response to 900 ppm CH_4_ exhibited only 2% signal attenuation (Figure 9b); (3) continuous operation under dry air flow at 2.1 V maintained baseline stability with zero drift (Figure 9c). This multi-faceted validation confirms outstanding operational repeatability and stability for long-term deployment.

Selectivity is a crucial indicator for sensors to achieve reliable detection in complex atmospheric environments. To this end, we selected various interfering gases, including CH_4_, C_2_H_6_, C_3_H_8_, O_2_, N_2_, and CO_2_, to evaluate the sensor’s selectivity under identical temperature and humidity conditions, measuring its response values to each gas type (Figure 9d). The test results demonstrate that the sensor exhibits a significantly higher response to alkane gases (CH_4_, C_2_H_6_, and C_3_H_8_) compared to other gases commonly present in atmospheric environments. This differential response among the alkanes arises from variations in their catalytic combustion characteristics, particularly the catalytic ignition temperature and associated heat release. These findings confirm the sensor’s excellent repeatability and specificity for methane and other alkane gases, highlighting its potential utility in various gas sensing applications.

### 3.8. Reaction Mechanism

The reaction mechanisms of catalytic combustion can be divided into two classical types: the Langmuir–Hinshelwood (L-H) mechanism and the Eley–Rideal (E-R) mechanism. The L-H mechanism requires both reactants to adsorb onto the catalyst surface before reaction, whereas the E-R mechanism allows one reactant to react directly from the gas phase with an adsorbed species.

Theoretical analysis indicates that the reaction mechanism of the catalytic combustion-type sensor prepared in this study follows the E-R mechanism: atmospheric O_2_, as a reactant with constant concentration, first adsorbs onto the active sites of the catalyst surface. Subsequently, O_2_ is dissociated by the Pd catalyst surface to provide oxygen atoms. The other reactant, CH_4_, does not require pre-adsorption but directly collides from the gas phase onto the catalyst surface, reacting with the dissociated oxygen atoms to release heat.

In this mechanism, the Pd catalyst functions as “targets” on the support surface: reactant CH_4_ is captured upon collision with the catalyst on the support surface and reacts with adsorbed oxygen atoms under Pd catalysis, after which the products desorb and leave the surface. Since CH_4_ does not need to occupy surface sites, its reaction rate is not limited by CH_4_ coverage on the support surface.

To further determine the catalytic combustion reaction mechanism, mathematical reasoning was employed to verify the E-R mechanism. The simplified kinetic form of the E-R mechanism under constant oxygen conditions aligns closely with the Langmuir equation. The chemical equation (Equation (10)) for methane–oxygen catalytic combustion is as follows:(10)$${\mathrm{CH}}_{4}(g)\;+\;2O_{2}(g)\text{ → }{\mathrm{CO}}_{2}(g)\;+\;2H_{2}O(g)\;∆H=-890.4\text{ kJ/mol}$$

The experimental methane concentration is C, the gas molar volume is V_m_, the heat released per mole of methane during complete catalytic combustion is ΔH, the support thermal transfer efficiency is п, the specific heat capacity of platinum wire is c, the approximate mass of the platinum wire bead inside the element is m, the platinum wire length is L, the cross-sectional area is S, the platinum wire temperature rise is ΔT, and its resistivity changes by Υ%. When the bridge voltage is E, the resistances of the black and white beads are R1 and R2, respectively, with compensation resistors R3 and R4 (R3 = R4). Theoretical derivation shows that the sensor’s differential voltage response satisfies Equation (11), which is structurally analogous to the Langmuir equation [31].(11)$$\Delta U=\frac{ab\cdot C}{1+b\cdot C}$$$$a=\frac{E}{2},\;b=\frac{N}{2R2},\;N=\frac{\Delta H\cdot l\cdot \Upsilon \%\cdot п}{S\cdot m\cdot c\cdot Vm}$$

The fitting results for methane concentration versus voltage difference in Section 3.3 exhibit high consistency (R^2^ > 0.99) with the aforementioned Langmuir equation. Given that the Langmuir equation aligns closely with the simplified kinetic form of the E-R mechanism under constant excess oxygen conditions, this indicates that the reaction mechanism of the catalytic combustion gas sensor prepared in this study is likely the E-R mechanism.

### 3.9. Comparison of Detection Limits

With a detection limit (LOD) of 36.14 ppm (3σ noise = 0.03 mV, Equation (4)), this sensor surpasses prior studies (Table 5) and critical safety standards, demonstrating 27.7% lower LOD than China’s GB 15322.1-2019 and ISO 26,142 (50 ppm) [32], and 63.9% below EU EN 60079-29-1 [33] and Japan’s JIS/Coal Mine Safety regulations (100 ppm) [34]. This unprecedented sensitivity to trace methane validates its deployment potential in industrial safety monitoring, residential leak detection, and environmental surveillance systems requiring early-warning capabilities at sub-LEL concentrations.

## 4. Conclusions

This study innovatively leveraged the unique channel structure and hydroxyl-rich surface of attapulgite to fabricate an attapulgite support with hierarchical pore architecture and strong metal anchoring capability. Through combined heat treatment, acid treatment, and palladium salt solution impregnation processes, a core–shell structured detection element was successfully synthesized. Within this configuration, palladium (Pd) exhibits a distinct concentration gradient from interior to exterior. EDS analysis confirms Pd concentration increases progressively from 7.5 wt% internally to 75.8 wt% at the outer surface. This architecture enriches Pd near the support periphery, enhancing gas–catalyst contact area and improving metal utilization efficiency. Performance evaluation under dynamic methane concentrations demonstrates ultra-high ppm-level sensitivity. The sensor achieves catalytic combustion under 300 °C, maintaining a theoretical signal output range of 0.03–41.80 mV with a sensitivity of 0.7 µV/ppm. The detection limit reaches 36 ppm, with reaction kinetics following the E-R mechanism. The voltage difference (ΔU) versus methane concentration (C) adheres to the Langmuir equation ($\Delta U=\frac{U_{max}\cdot K\cdot C}{1+K\cdot C}$, R^2^ > 0.99, $U_{max}\;$= 41.80 mV, K = 2 ×10^−5^). Owing to its cost-effective fabrication and exceptional sensitivity, this sensor exhibits significant application potential in industrial/commercial safety monitoring, household security systems, and environmental surveillance.

## Figures and Tables

**Figure 1 sensors-25-04950-f001:**
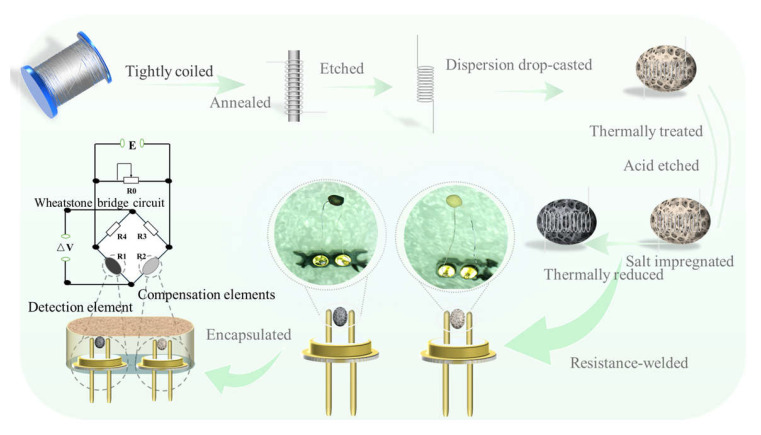
Preparation flowchart of catalytic combustion gas sensor (Wheatstone bridge circuit).

**Figure 2 sensors-25-04950-f002:**
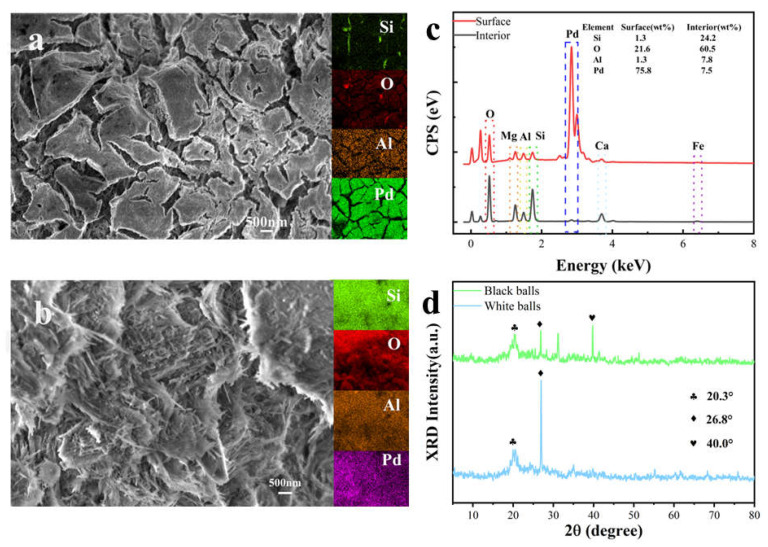
(**a**) Surface morphology (SEM) and corresponding EDS elemental mapping (Si, O, Al, and Pd) of the black bead; (**b**) internal morphology (SEM) and corresponding EDS elemental mapping (Si, O, Al, and Pd) of the black bead; (**c**) comparative EDS spectra of black bead surface vs. interior; and (**d**) XRD patterns of black and white beads.

**Figure 3 sensors-25-04950-f003:**
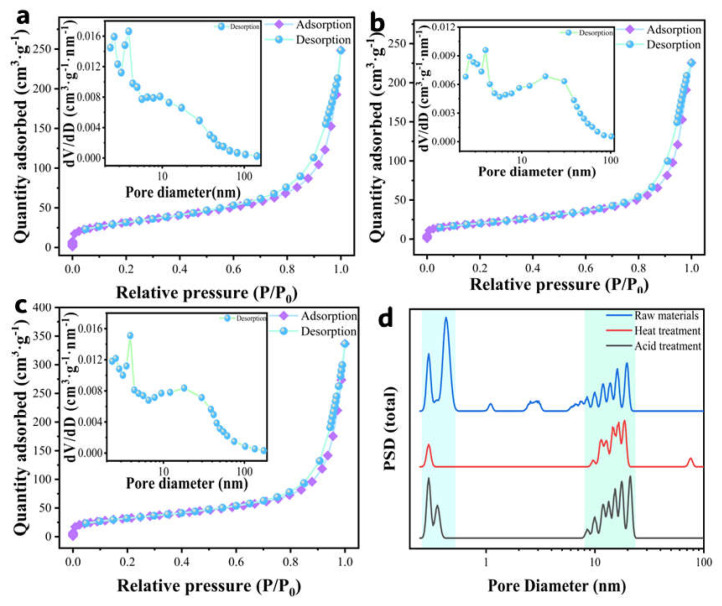
Nitrogen adsorption curve and pore distribution of white sphere (**a**) raw material; (**b**) heat treatment; (**c**) acid treatment; and (**d**) pore volume distribution.

**Figure 4 sensors-25-04950-f004:**
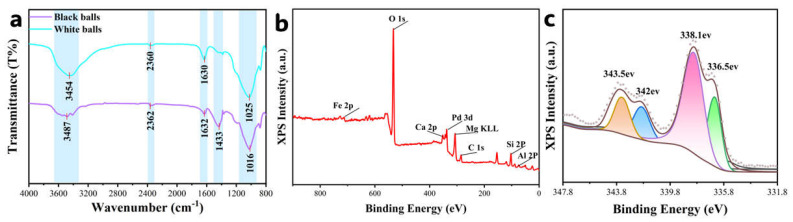
(**a**) FTIR spectra of black and white beads; (**b**) XPS full spectrum of the black bead surface; and (**c**) high-resolution Pd 3d scan of the black bead surface.

**Figure 5 sensors-25-04950-f005:**
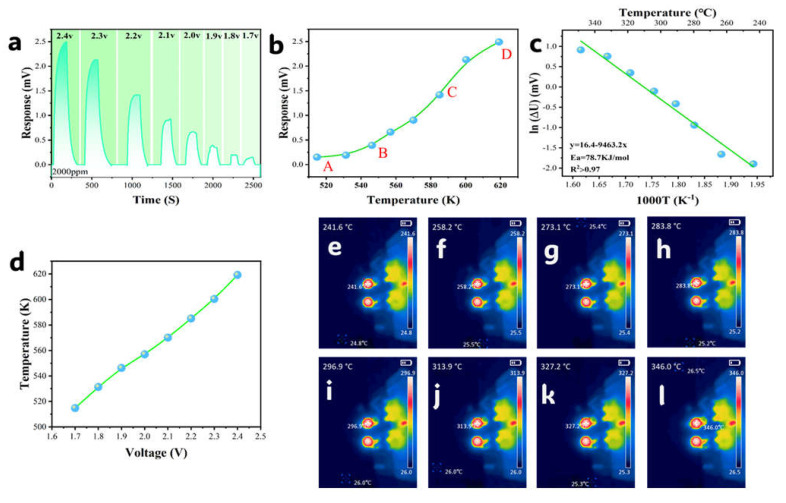
(**a**) Differential response vs. operating voltage at 2000 ppm CH_4_; (**b**) differential response vs. operating temperature; (**c**) Arrhenius fitting of temperature-dependent response; (**d**) sensor surface temperature vs. operating voltage; and (**e**–**l**) thermal imaging at operating voltages (1.7–2.4 V).

**Figure 6 sensors-25-04950-f006:**
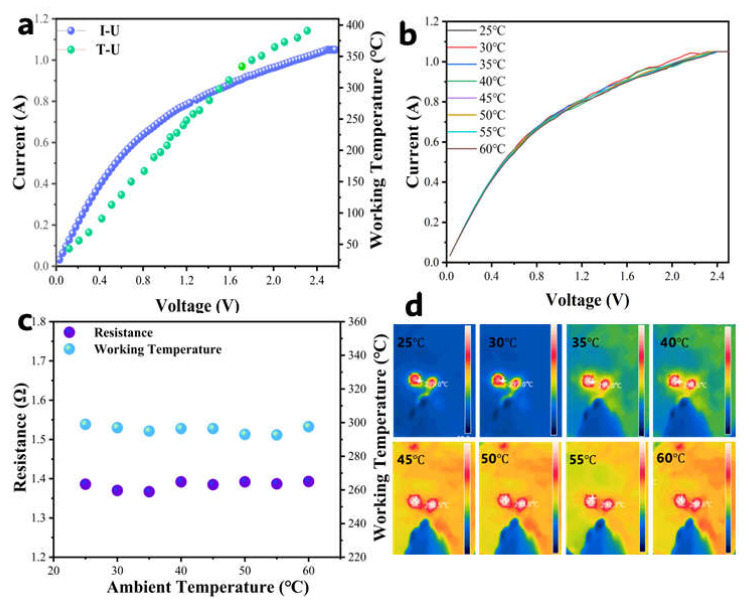
(**a**) The voltage–current/working temperature relationship for the detection element (black bead) measured at 25 °C (ambient temperature); (**b**) the voltage–current relationship for the detection element (black bead) measured under varying ambient temperature (25–60 °C); and (**c**,**d**) the resistance and working temperature of the detection element (black bead) under the stable voltage (1.05 V) across ambient temperatures range from 25 to 60 °C.

**Figure 7 sensors-25-04950-f007:**
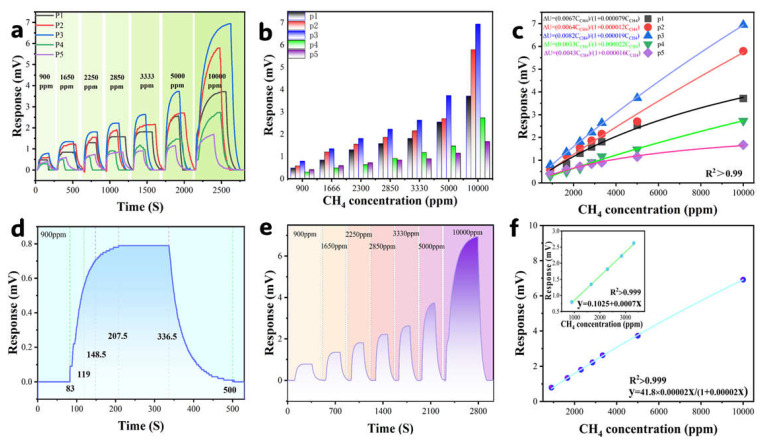
(**a**) Differential voltage response vs. CH_4_ concentration at varied Pd loadings; (**b**) response comparison across Pd loadings; (**c**) Langmuir fitting of dynamic responses; (**d**) a complete response–recovery cycle at 2000 ppm methane concentration; (**e**) differential voltage response of the P3 sensor versus methane concentration; and (**f**) fitting curve of the P3 sensor’s differential voltage response versus methane concentration.

**Figure 8 sensors-25-04950-f008:**
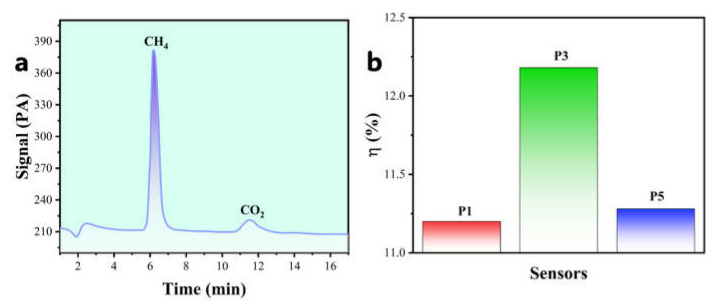
(**a**) Gas chromatography (GC) analysis; (**b**) catalytic efficiency.

**Figure 9 sensors-25-04950-f009:**
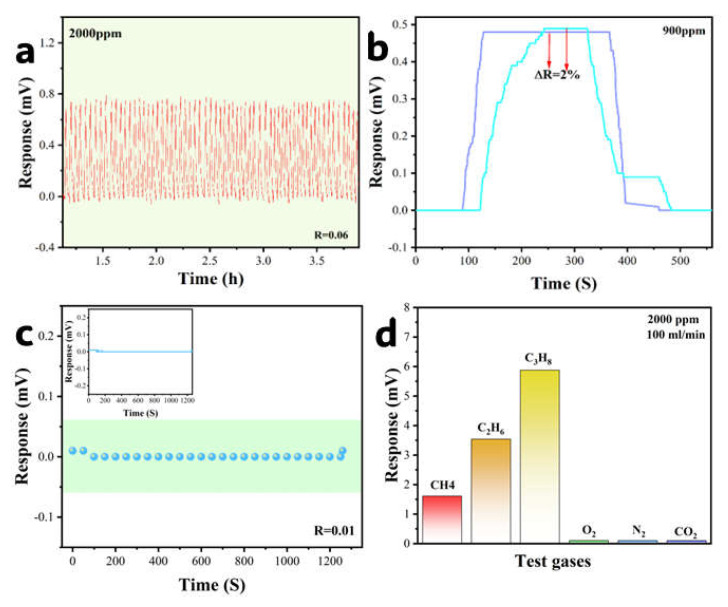
(**a**) 100-cycle repeatability test at 2000 ppm CH_4_; (**b**) response consistency over 40-day intervals; (**c**) stability during continuous standby operation; (**d**) selectivity response for various interfering gases of the sensor.

**Table 1 sensors-25-04950-t001:** Physical properties of attapulgite support after heat treatment and acid treatment.

Samples	S_BET_ (m^2^·g^−1^)	Pore Volume (cm^3^·g^−1^)	Average Pore Size (nm)	Main Pore Diameter (nm)
Raw materials	138.98	0.3333	6.75	0.63
Heat treatment	91.39	0.3943	9.51	18.72
Acid treatment	139.98	0.4753	9.43	21.12

**Table 2 sensors-25-04950-t002:** Comparative FTIR analysis of functional groups on black and white beads.

	3454 cm^−1^ (-OH)	2360 cm^−1^ (O=C=O)	1630 cm^−1^ (H-O-H)	1433 cm^−1^ (CO_3_^2−^)	1025 cm^−1^ (Si-O)
Black	weaken	exist	exist	exist	exist
White	exist	exist	exist	absent	right shift

**Table 3 sensors-25-04950-t003:** Response kinetics parameters for P1-P5 sensors with varying Pd loadings.

SAMPLE	U_90_ (mV)	T_90_ (S)	△U/△T (µV/S)	X (%)
P1	3.339	177.5	17.8	61.59
P2	5.011	209.5	23.9	82.70
P3	6.246	215.5	28.9	100
P4	2.457	202.5	12.7	43.94
P5	1.053	126	11.9	41.18

**Table 4 sensors-25-04950-t004:** Methane catalytic combustion efficiency of P1, P3, and P5 sensors.

	Wt%	$S_{{CH}_{4}}$	$S_{{CO}_{2}}$	η
P1	2	5937.95	748.62	11.20
P3	3	5469.33	758.338	12.18
P5	4	5268.34	670.27	11.28

**Table 5 sensors-25-04950-t005:** Examples of preliminary work for catalytic combustion gas sensors.

Sensing Material	Response (%)	τ Res/τ Recov (s)	LOD (ppm)	Working Temperature/°C	Ref.
VO_2_-MoTe_2_	13 (500 ppm)	75/110	500	RT	[35]
GaN	4.5 (500 ppm)	126/132	100	200	[36]
ZnO	62 (1000 ppm)	190/204	100	RT	[37]
V_2_O_5_	11.2 (500 ppm)	230/187	50	100	[38]
Pt-Co_3_O_4_/MoS_2_	21 (500 ppm)	30/25	100	180	[39]
NIO/rGO	15.2 (1000 ppm)	16/20	500	350	[40]
Pt–Ca/SnO_2_	2.3 (5000 ppm)	_	_	400	[41]
Fe–SnO_2_	70 (1000 ppm)	_	_	350	[42]
SnO_2_ NRS-NPG	24.9 (1000 ppm)	369/_	1000	600	[43]
Pd–SnO_2_	97.2 (200 ppm)	26/70	_	220	[44]
ZnO–SnO_2_	80 (5000 ppm)	_	_	350	[45]
Ni_2_O_3_–SnO_2_	127 (200 ppm)	_	_	400	[46]
Ag–SnO_2_	75 (2000 ppm)	_	_	430	[47]
Pd/SnO_2_-rGO	9.5 (12,000 ppm)	300/420	_	RT	[48]
Pt–SnO_2_	4.5 (1000 ppm)	25/141	_	300	[49]
Pd/attapulgite	107 (900 ppm)	65.5/163.5	36	296	This work

## Data Availability

Data are contained within the article.

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
