# Peer review of "Pd/Attapulgite Core–Shell Structured Catalytic Combustion Gas Sensor for Highly Sensitive Real-Time Methane Detection"

_sensors, 2025, doi:10.3390/s25164950_

Round 1

Reviewer 1 Report

Comments and Suggestions for Authors

Cao et al. showed a catalytic combustion gas sensor based on Pd/Attapulgite(Mg, Al)2Si4O10(OH)catalyst for sensing of methane. Overall, the manuscript is clearly written. However, the presented paper does not provide sufficient data or new insights in term of sensor science to warrant a publication in Sensors. If the authors want to study on the effect of the catalytic material, the experiments of more subjects of different material samples should be designed.

The catalyst in this manuscript is almost same to the previous report of authors group, Journal of Chromatography B 1247, 124338 (2024), and Journal of Environmental Chemical Engineering 13, 117528 (2025). What is important in this manuscript is to understand the combustion performance of the catalyst, but their discussion is not supported by clear experimental evidence. Discussing it based on the Langmuir adsorption model is essentially meaningless for sensor performaces. The authors should thoroughly discuss the combustion model. For example, by checking the temperature dependency, it is possible to determine which reaction is rate-limiting. When using a catalytic combustion type sensor with a reaction-limiting reaction, there is a risk that the gas concentration of the target object will not be measured correctly. The authors need to take this into consideration and reconsider his data and experimental method.

For the Pt-coil-type sensors are not new, so there is nothing novel about them. Newer technologies should be introduced and compared, such thermoelectric types, thermoelectric (TE) gas sensor, E. Vereshchagina et al 2015, Sensors and Actuators B 206, 772-787. For the kinetics and related discussions on catalytic combustion and D. Nagai et al 2014 Sensors 14, 1822-1834, S. Lee et al 2003 Microelectron. J. 34, 115-126, would be informative for the authors.

The combustion characteristics of sensors P1 to P5 are compared, but why do these differences occur? What about the microstructure of the catalyst? What about the issues of combustion speed and heat balance? Are the catalytic activities different? Various discussions are needed, but they are not mentioned in the text.

Author Response

We sincerely appreciate the time you dedicated to reviewing our manuscript and are deeply grateful for your invaluable comments. Your insightful feedback has immensely helped us not only significantly improve the quality of this paper but also provided valuable inspiration for our future research directions. During the revision process, we supplemented multiple critical experiments in response to your feedback:

  1. Evaluated the stabilityof the working temperature using infrared thermography across ambient temperatures (25-60°C).
  2. Quantified response ratethrough T₉₀ analysis.
  3. Characterized gas reaction pathways via gas chromatography with compositional quantification.

Furthermore, substantive discussions have been incorporated to strengthen the manuscript's scholarly rigor and interpretative depth.

In response to your expert feedback, we have thoroughly revised the manuscript and ensured a point-by-point response to each of your suggestions. Please see the attachment.

Reviewer 2 Report

Comments and Suggestions for Authors

This manuscript (sensors-3763872) reported a methane sensor using Pd/attapulgite core-shell structured catalytic as the gas sensitive material. The results of methane gas sensor are basically acceptable, but the writing, experimental details, and discussions need to be improved. The manuscript needs a revision before possible acceptance.

  1. …at low temperatures (< 300 °C). 300 °C cannot be considered as low temperature.
  2. The content from the third to the fourth paragraph of the introduction fails to clearly establish internal logic and lacks persuasiveness regarding the necessity of studying attapulgite. In addition, compared with other methane gas-sensitive materials, such as precious metals, metal oxides, and the ceramic materials mentioned in the third paragraph, what are the advantages of the gas-sensitive material studied in this paper? Moreover, the rationale for choosing attapulgite combined with Pd to form a core-shell structure as a gas-sensitive material is not sufficiently convincing. Attapulgite have actually been reported in the field of sensors, such as humidity sensors (Sens. Actuators B Chem., 2020, 305, 127534). In addition, the natural nanomaterials similar to attapulgite, such as halloysite nanotubes, have also been reported for the preparation of gas sensors (PANI/halloysite nanotubes). It is suggested to discuss this research status to highlight research motivation and innovation.
  3. Experimental section: Material sources need to be provided.
  4. Figure 2: How the characterization samples were prepared.
  5. Please elaborate on the calculation methods for response/recovery time, response, and sensitivity of the methane sensor.
  6. How does the optimal methane sensor studied in this paper perform in terms of gas selectivity?
  7. Table 3: The number of references compared is too small, at least 10.
  8. The sensing materials in the last row of Table 3 are inconsistent with those studied in this paper.
  9. The line width and font size of the insets in the figures need to be improved.
  10. Check the reference format. Abbreviating the journal name. The numbers in the chemical formula require subscripts, including references.
  11. Check English writing.
Comments on the Quality of English Language

Check English writing

Author Response

We sincerely appreciate the time you dedicated to reviewing our manuscript and are deeply grateful for your invaluable comments. Your insightful feedback has immensely helped us not only significantly improve the quality of this paper but also provided valuable inspiration for our future research directions. We are truly honored that your suggestions have enabled our team to gain a deeper understanding of the research in this field. In response to your expert feedback, we have thoroughly revised the manuscript and ensured a point-by-point response to each of your suggestions. Please see the attachment.

Round 2

Reviewer 1 Report

Comments and Suggestions for Authors

The explanation of from P1 to P5 is still vague and of lacks an explanation of the composition or microstructure. It is clearly shown that the rate-limiting reaction is not applicable to the reaction of sensor P4 or P5, but the differences between each sensor regarding these differences are not explained. Without these explanations or discussion, this paper would not state any new scientific discoveries.

The discussion of the paper is insufficient in many places, and many of them seem to have been written in a hurry. For example, the captions of Table 3 and Table 4 are difficult to understand even after reading them.

I have introduced a paper (D. Nagai et al 2014 Sensors) because I thought it was necessary to discuss heat balance. However, the authors just add ref as MEMS device, which simply increases the ref number. In this paper the authors did not carefully measures the catalyst characteristics, so that it cannot be a paper of catalysts, but as a paper on sensors, the conditions under which the temperature of the sensor element is proportional to the gas concentration through the balance between heat generation and heat loss are important. There must be results and discussions on the various changes depending on the microstructure of the associated catalyst, and in some cases the size.

Author Response

We sincerely thank you for the time and effort you have invested in reviewing the manuscript, as well as for your valuable suggestions. Your profound advice has not only significantly enhanced the quality of this paper but also provided important inspiration for our future research directions, further deepening our understanding of this field.

Specifically, the structure-activity relationship analysis is indeed the key of this study. To this end, we systematically analyzed the surface microstructure of the sensing elements (black spheres) in P1, P3, and P5 devices by combining scanning electron microscopy (SEM), and drew relevant conclusions accordingly. Furthermore, substantive discussions about heat balance have been incorporated to strengthen the manuscript's scholarly rigor and interpretative depth.

In response to your expert feedback, we have thoroughly revised the manuscript and ensured a point-by-point response to each of your suggestions. Please see the attachment.

Reviewer 2 Report

Comments and Suggestions for Authors

The response and revised manuscript are satisfactory, and it is recommended to accept.

Author Response

Thank you very much for your time and comments.

Round 3

Reviewer 1 Report

Comments and Suggestions for Authors

The authors revised the manuscript responding most critical issues to my advices and suggestions.